# Differential Inflammatory Responses in Cultured Endothelial Cells Exposed to Two Conjugated Linoleic Acids (CLAs) under a Pro-Inflammatory Condition

**DOI:** 10.3390/ijms23116101

**Published:** 2022-05-29

**Authors:** Carina A. Valenzuela, Ella J. Baker, Elizabeth A. Miles, Philip C. Calder

**Affiliations:** 1School of Human Development and Health, Faculty of Medicine, University of Southampton, Southampton SO16 6YD, UK; e.baker@soton.ac.uk (E.J.B.); e.a.miles@soton.ac.uk (E.A.M.); pcc@soton.ac.uk (P.C.C.); 2Escuela de Nutrición y Dietética, Facultad de Farmacia, Universidad de Valparaíso, Playa Ancha, Valparaíso 2360102, Chile; 3NIHR Southampton Biomedical Research Centre, University Hospital Southampton NHS Foundation Trust, University of Southampton, Southampton SO16 6YD, UK

**Keywords:** conjugated fatty acids, inflammation, endothelial cells, atherosclerosis

## Abstract

Conjugated linoleic acid (CLA) isomers have been shown to possess anti-atherosclerotic properties, which may be related to the downregulation of inflammatory pathways in different cell types, including endothelial cells (ECs). However, whether different CLA isomers have different actions is not entirely clear, with inconsistent reports to date. Furthermore, in cell culture studies, CLAs have often been used at fairly high concentrations. Whether lower concentrations of CLAs are able to affect EC responses is not clear. The aim of this study was to evaluate the effects of two CLAs (*cis*-9, *trans*-11 (CLA9,11) and *trans*-10, *cis*-12 (CLA10,12)) on the inflammatory responses of ECs. ECs (EA.hy926 cells) were cultured under standard conditions and exposed to CLAs (1 to 50 μM) for 48 h. Then, the cells were cultured for a further 6 or 24 h with tumour necrosis factor alpha (TNF-α, 1 ng/mL) as an inflammatory stimulant. ECs remained viable after treatments with 1 and 10 μM of each CLA, but not after treatment with 50 μM of CLA10,12. CLAs were incorporated into ECs in a concentration-dependent manner. CLA10,12 increased the levels of ICAM-1, IL-6, and RANTES in the culture medium, while CLA9,11 had null effects. Both CLAs (1 μM) decreased the appearance of NFκB1 mRNA, but only CLA9,11 maintained this downregulation at 10 μM. CLA10,12 had no effect on THP-1 cell adhesion to ECs while significantly decreasing the percentage of ECs expressing ICAM-1 and also levels of ICAM-1 expression per cell when used at 10 µM. Although CLA9,11 did not have any effect on ICAM-1 cell surface expression, it reduced THP-1 cell adhesion to the EA.hy926 cell monolayer at both concentrations. In summary, CLA10,12 showed some pro-inflammatory effects, while CLA9,11 exhibited null or anti-inflammatory effects. The results suggest that each CLA has different effects in ECs under a pro-inflammatory condition, highlighting the need to evaluate the effects of CLA isomers independently.

## 1. Introduction

Conjugated linoleic acids (CLAs) are a mixture of positional and geometric isomers of linoleic acid. The main CLA isomer found in the human diet is *cis*-9, *trans*-11 linoleic acid (CLA9,11). CLA9,11, together with several other CLA isomers, is produced naturally in the rumen of cattle and other ruminants by bacterial biohydrogenation of linoleic acid [1]. The milk, dairy products, and meat of these animals are a source of CLAs, with CLA9,11 accounting for 70–80% of the total CLA content in dairy and meat products [2,3]. Consequently, CLA9,11 is the main CLA present in the human diet. It has been reported that humans can also endogenously produce CLA9,11 through desaturation of *trans* vaccenic acid, the main ruminant *trans* fatty acid [4,5]. *Trans*-10, *cis*-12 linoleic acid (CLA10,12) is a more minor constituent of foods than CLA9,11. However, CLA supplements, which are mainly produced from sunflower oil, which is rich in linoleic acid, typically contain a 50:50 mixture of CLA9,11 and CLA10,12. Both these CLA isomers are bioactive, and since the 1980s, different studies have frequently shown positive effects of CLAs in cancer, inflammation, obesity, glucose homeostasis, and atherosclerosis, among others. These studies are mainly performed in vitro and in animal models [6,7,8,9,10]. However, the effects on CLA isomers, alone or together, in humans are inconsistent, and the mechanisms of action are still unclear [2,11].

Atherosclerosis is the disease that results from the formation of atheromatous plaques within the arterial wall, causing thickening or hardening [12]. It plays a key role in the progression of cardiovascular diseases, including coronary artery disease and stroke, which are still among the leading causes of death globally [13]. Inflammation of the vascular endothelium is an important contributor to the initiation and progression of atherosclerosis [14,15,16].

The vascular endothelium can be modulated by components of the diet, such as fatty acids, and this modulation has implications for vascular-related diseases such as atherosclerosis. Some in vitro studies have reported anti-inflammatory effects of CLAs on endothelial cells (ECs), such as the commonly used human umbilical vein endothelial cells (HUVECs) [17,18,19]. In contrast, studies in mice show that dietary CLA10,12 increases the expression and circulating levels of inflammatory mediators and macrophage infiltration into adipose tissue [20,21]. Clinical trials also show inconsistent results, with some studies reporting that CLA10,12 may have pro-inflammatory effects compared to CLA9,11 [22,23,24,25], while other studies show no effects and no differences between the CLAs [26,27].

There is some evidence suggesting that CLA9,11 and CLA10,12 have different effects on inflammation, but this is uncertain, and in vitro studies have used high concentrations of the CLAs. Whether lower concentrations of CLAs are able to affect the inflammatory responses of ECs is not clear. The aim of this study was to evaluate the effects of two CLAs (CLA9,11 and CLA10,12) used at lower concentrations than in many previous in vitro studies on inflammatory responses by cultured ECs.

## 2. Results

### 2.1. Viability of EA.hy926 Cells Incubated with TNF-α and FAs

Viability was assessed using the 3-(4,5-dimethylthiazol-2-yl)-2,5-diphenyltetrazolium bromide (MTT) assay, which assesses metabolic integrity. Neither tumour necrosis factor (TNF)-α (1 ng/mL for 24 h) nor any of the three fatty acids (FAs) tested at concentrations of 1 and 10 µM affected the viability of EA.hy926 cells (Figure 1). However, at a concentration of 50 µM, CLA10,12 reduced viability. Therefore, further experiments did not use FAs at concentrations above 10 µM.

### 2.2. FA Incorporation into EA.hy926 Cells

The incorporation of each of the studied FAs into EA.hy926 cells increased as their concentration in the culture medium increased from 1 to 10 µM (Figure 2). CLA10,12 was incorporated in a higher amount (at least 50% more) than CLA9,11.

### 2.3. Effects of FAs on the Levels of Inflammatory Mediators Produced by ECs

Pre-incubation of EA.hy926 cells with any of the FAs tested prior to TNF-α stimulation did not induce changes in the supernatant levels of monocyte chemoattractant protein (MCP)-1 compared to control (i.e., TNF-α without any FA pre-incubation) (Figure 3A). 

Pre-incubation with CLA10,12 at 1 µM produced a significant increase (*p* < 0.01) in the supernatant levels of intercellular adhesion molecule (ICAM)-1 compared to control (Figure 3B).

Pre-incubation with linoleic acid (LA) at 1 µM decreased supernatant interleukin (IL)-6 levels, behaving significantly different from CLA10,12, while at 10 µM, CLA10,12 produced a significant increase in IL-6 levels in the supernatant compared to control, LA and CLA9,11 (*p* < 0.0001, *p* < 0.001, and *p* < 0.0001, respectively) (Figure 3C).

CLA pre-treatments did not induce significant changes in supernatant IL-8 levels compared to control, although pre-treatment with LA at 1 µM resulted in a significant increase in IL-8 levels in the supernatant compared to control and CLA10,12 (both *p* < 0.05) (Figure 3D).

For regulation upon activation, normal T cells expressed and presumably secreted (RANTES), pre-incubation with CLA10,12 at both concentrations induced a significant increase compared to control (*p* < 0.0001 and *p* < 0.001, respectively), behaving differently from LA and CLA9,11 (Figure 3E).

### 2.4. Effects of FAs on the Expression of Inflammation-Related Genes

Pre-incubation with both CLAs at 1 µM induced a significant reduction in the relative expression of the nuclear factor kappa-light-chain-enhancer of activated cells 1 (NFκB1) gene (*NFκB1*) compared to control (*p* < 0.01), acting differently from LA (*p* < 0.05 and *p* < 0.01, respectively) (Figure 4A). When used at 10 µM, only CLA9,11 sustained this effect (*p* < 0.01).

None of the FAs used, at either concentration, induced significant changes in the relative expression of the inhibitor subunit of kappa B α (IκBα) gene (this gene is termed *NF**κ**BIA*) in comparison to the control. However, when used at 1 µM, LA behaved significantly differently from CLA10,12 (Figure 4B).

Pre-incubation with CLA10,12 increased expression of the IκB kinase subunit B (IκK-β) gene (this gene is termed *I**κ**BKB*) when used at 1 µM. LA had the opposite effect at both concentrations used, decreasing the expression of the IκK-β gene. CLA9,11 had no effect at either concentration (Figure 4C).

Pre-incubation with the FAs tested did not induce significant changes in peroxisome proliferator activated receptor (PPAR)-α gene expression following TNF-α stimulation, although CLA10,12 tended to increase it, behaving significantly differently than CLA9,11 and LA at 10 µM (Figure 4D).

Both CLAs at 1 and 10 µM, but particularly CLA9,11, tended to increase expression of *PTGS2*, the gene encoding cyclooxygenase 2 (COX-2), although the effects were not statistically significant (Figure 4E).

Pre-incubation with the CLAs did not induce changes in IL-6 gene expression (Figure 4F).

### 2.5. Effects of FAs on THP-1 Adhesion to EA.hy926 Cells

Pre-incubation of EA.hy926 cells with either LA or CLA9,11 at 1 µM decreased adhesion of THP-1 cells compared to stimulated control cells (*p* < 0.05) (Figure 5A), with a more pronounced effect when used at 10 µM (Figure 5B, *p* < 0.0001 and *p* < 0.01, respectively). CLA10,12 had no effect on THP-1 adhesion.

Images taken under the fluorescence microscope agree with the quantitative results, showing a lower number of THP-1 monocytes (green spots) when ECs were pre-incubated with LA and CLA9,11 (Figure 6B,C).

### 2.6. Effects of FAs on the Expression of ICAM-1 on the Surface of EA.hy926 Cells

Incubation of EA.hy926 cells with TNF-α significantly up-regulated cell surface ICAM-1 expression [28].

Pre-incubation with CLA9,11 at either 1 or 10 µM produced no change in either the percentage of ECs expressing ICAM-1 or in the levels of ICAM-1 expression per cell (Figure 7), behaving similar to control. CLA10,12 induced an increase in the percentage of cells expressing ICAM-1 when used at 1 µM (Figure 7A, *p* < 0.01), behaving significantly differently from CLA9,11 and LA at that concentration (*p* < 0.001) while inducing a significant reduction both in the percentage of ECs expressing ICAM-1 and in the levels of ICAM-1 expression per cell when used at 10 µM (Figure 7B,D), *p* < 0.05 and *p* < 0.01, respectively).

## 3. Discussion

This study examined the effects of CLA9,11 and CLA10,12 at concentrations of 1 and 10 μM on inflammatory responses of ECs in comparison to their all-*cis*-isomer, LA, and a control with no additional FA. The two CLA isomers were shown to have different effects on EC responses to an inflammatory stimulant (TNF-α) when they were used at these low concentrations. CLA10,12 had a number of pro-inflammatory effects, while CLA9,11 had either null or anti-inflammatory effects. CLAs were dissolved in ethanol prior to addition to cultures as follows: the final ethanol concentration in all cultures was 0.1%. This concentration of ethanol had no effect on any of the outcomes reported in this study (data not shown).

The involvement of different *trans* fatty acids, including *trans* vaccenic acid and CLA isomers, in the modulation of inflammatory processes is not fully understood [29,30]. In the current study, CLA10,12 increased the concentrations of ICAM-1, RANTES and IL-6 in the supernatant of ECs after TNF-α stimulation, suggesting that CLA10,12 increases the production of those inflammatory mediators. CLA9,11 had no effect on any of the inflammatory mediators measured. In terms of gene expression, both CLAs decreased the expression of *NFκB1* after TNF-α stimulation when used at 1 μM, but only CLA9,11 maintained this downregulation at 10 μM. This could be considered an anti-inflammatory effect if decreased gene expression resulted in less NFκB protein being produced. Paradoxically, at a concentration of 1 µM, CLA10,12 also induced increased expression of the gene encoding IκKβ, which may lead to the activation of the NFκB pathway if it translates into more IκKβ protein being produced. The levels of these proteins were not assessed in the current study. CLA10,12 pre-treatment of ECs had no effect on THP-1 adhesion although it significantly decreased the percentage of ECs expressing the adhesion molecule ICAM-1 and also levels of ICAM-1 expression per cell when used at 10 µM. This observation suggests that EC ICAM-1 is not involved in the adhesion of monocytes. The observations that CLA10,12 promotes some inflammatory responses but not others suggest that CLA10,12 has a complex modulatory role in the inflammatory response of ECs, probably through different and maybe opposing mechanisms. The lower expression of ICAM-1 on the EC surface after treatment with CLA10,12 may be linked to the higher concentration of (soluble) ICAM-1 observed in the culture medium of those cells. CLA10,12 may act to promote cleavage of ICAM-1 from the cell surface, meaning that ICAM-1 is released from the cell surface into the culture medium.

In the case of CLA9,11, some anti-inflammatory effects were observed, including decreased *NF**κB1* expression and decreased adhesion of THP-1 cells to ECs. CLA9,11 did not affect ICAM-1 expression on ECs. Since CLA9,11 decreased adhesion but not ICAM-1 expression, this again indicates that ICAM-1 is not involved in THP-1 binding to ECs. The effect of CLA9,11 on the expression of other candidate adhesion molecules such as vascular cell adhesion molecule 1 (VCAM-1) should be explored in future research. 

Differential effects of CLA isomers have also been reported by others. In healthy postmenopausal women, supplementation with CLA10,12 caused higher plasma levels of C-reactive protein (CRP), fibrinogen, and plasminogen activator inhibitor-1 (PAI-1) and higher concentrations of a urinary marker of lipid peroxidation compared to CLA9,11 and the olive oil, although, plasma levels of IL-6, sVCAM-1, sICAM-1, MCP-1, and TNF-α were not different between the groups [22]. Other studies have also reported increased CRP after CLA10,12 or a mix of CLAs, including CLA10,12, in obese men with metabolic syndrome [23], obese adults [24], or healthy adults [25]. Nevertheless, the evidence for pro-inflammatory effects of CLA10,12 in humans is not consistent. A supplementation study in healthy young adults with CLA9,11 or CLA10,12 for 8 weeks showed that neither of the isomers affected lymphocyte subpopulations, serum concentrations of CRP or ex vivo cytokine production by peripheral blood mononuclear cells (PBMCs) in response to different inflammatory stimuli [31,32]. Only some differences in blood lipids and the expression of ICAM-1 on monocytes were observed [31,32]. In the latter study, there was a suggestion of fewer ICAM-positive monocytes in the blood after CLA10,12, which parallels the effects seen in the current study for CLA10,12 and EC ICAM-1. Ramakers et al. found no effects on ex vivo cytokine production by isolated PBMCs or by PBMCs present in whole blood when stimulated with lipopolysaccharide (LPS) from a small sample of moderately overweight subjects at increased risk for coronary artery disease, after daily consumption of 3 g of CLA9,11 or CLA10,12 in an enriched dairy product for 13 weeks [26]. Another study testing a CLA-enriched butter for 5 weeks showed increases in lipid peroxidation but no effects on plasma total-, low denity lipoprotein-, and high density lipoprotein-cholesterol and triglycerides, or inflammatory and haemostatic risk markers, nor in fasting insulin and glucose concentrations in healthy young men [33]. 

In an animal model of apolipoprotein E knockout (ApoE^−/−^) mice, the development of atherosclerotic lesions was reduced by a CLA9,11 supplemented diet, while a CLA10,12 supplemented diet resulted in pro-atherogenic effects [34]. The authors also reported that the mice fed CLA9,11 had stabilised plaques with a higher content of smooth muscle cells and collagen than the CLA10,12 fed group. In the animals fed with CLA10,12, the fibrous cap was replaced by an acellular mass with higher macrophage content and activation underneath the plaques. The authors suggested that the two isomers displayed a different capacity to repress or accelerate the progress of atherosclerosis in that animal model [34]. It appears from the observations that these effects were based around inflammation. In contrast, a recent study in ApoE^−/−^ mice fed with a high fat and high cholesterol diet showed that an 80/20 CLA blend (CLA9,11 and CLA10,12, respectively) limited atherosclerosis progression by promoting an increase in the anti-inflammatory M2 macrophage phenotype [35]. The high proportion of CLA9,11 vs. CLA10,12 resembles the ratio in which they are naturally produced.

In C57BL/6J female mice, the administration of CLA10,12 by gavage at a dose of 20 mg/day for 7 days led to the upregulation of *TNF-α*, *MCP-1* and *IL-6* expression in white adipose tissue (WAT) without affecting their plasma levels, together with macrophage infiltration into WAT, upregulation of *suppressor of cytokine signalling 3* (*SOCS3*) and downregulation of *PPAR-γ* expression in WAT [21]. Another study, using diets enriched with a mix of CLA10,12 with LA (50/50), a mix of CLA10,12 with CLA9,11 (50/50), or LA alone as a control in young male mice for 6 weeks, showed that the intermediate and higher intakes of CLA10,12 reduced adiposity, increased plasma levels of MCP-1 and IL-6, and increased liver steatosis [20]. Again, these animal studies strongly suggest that CLA10,12 has pro-inflammatory effects, as seen in the current study. Some in vitro studies also agree with this conclusion. Goua et al. [14] showed that CLA10,12 and a CLA mix (CLA10,12 and CLA9,11) reduced ICAM-1 and VCAM-1 expression on HUVECs, although only the CLA mix (with CLA9,11) was used at 25 µM was able to decrease NFκB activity (by 30%) in both HUVECs and smooth muscle cells treated with TNF-α. Poirier et al. [17] showed that when 3T3-L1 adipocytes were exposed to CLA10,12, there was increased nuclear localization of the p65 subunit of NFκB, upregulation of *IL-6* expression and IL-6 secretion, downregulation of the *PPAR-γ* expression and PPAR-γ protein and upregulation of *SOCS3*. Once again, this study suggests the pro-inflammatory effects of CLA10,12.

In terms of the mechanism of action of the CLAs, it seems feasible that the NFκB pathway is involved, together with PPAR-α and -γ. CLA10,12 tended to upregulate *PPAR-α* expression and decreased *NFκB1* expression in the current study. It is known that PPAR-α can negatively regulate pro-inflammatory gene expression by antagonising the activities of other transcription factors, including NFκB [36]. However, these effects on the NFκB or PPAR-α pathways seem unlikely to explain how CLA10,12 resulted in increased production of ICAM-1, RANTES, and IL-6. CLA9,11 also decreased *NF**κB1* expression, although this was not linked with decreased production of inflammatory cytokines. It might, however, be linked to the reduced adhesion of THP-1 monocytes, especially if the altered expression of one or more unidentified adhesion molecules was involved in that effect.

FAs can affect inflammatory processes through multiple candidate mechanisms [37,38]. They can act directly through the PPAR system since PPARs are cytosolic FA receptors, and this mechanism of action has been reported for some CLAs [39,40]. As described earlier, PPARs can antagonize NFκB [36]. However, FAs can also influence the signalling pathway that activates NFκB; this is most well described for omega-3 FAs. Eicosapentaenoic acid and docosahexaenoic acid (DHA) are both described to inhibit activation of NFκB in several cell types, including dendritic cells [41,42], monocytes [43,44], macrophages [45], and ECs [46], so decreasing its nuclear translocation. Detailed studies with DHA in dendritic cells and macrophages identified that the inhibition of NFκB activation in response to LPS, which involved decreased phosphorylation of IκB, stemmed from membrane-mediated events [41,42,45]. DHA was shown to prevent the formation of lipid rafts in the plasma membrane in response to LPS stimulation [42,45], the result of which was an inability to assemble the earliest signalling platforms that ultimately activate IκBK and NFκB. This research indicates that actions exerted in the plasma membrane, ultimately, can have their effects on the nucleus. DHA seems to be quite potent as a raft disrupter because of its unique physical structure that results from its long carbon chain (22 carbons) and a high degree of unsaturation (6 double bonds) [47]. Hence, the incorporation of DHA into the plasma membrane induces physical changes that influence membrane-mediated responses to cell stimulation. Interestingly, some saturated fatty acids seem to have the opposite effect in macrophages. where they act to promote raft formation in response to LPS, thus increasing inflammatory responses [42,45]. By definition, CLAs have an unusual structural feature, the presence of a conjugated double bond. This may give CLAs membrane modifying properties, with the two CLAs (and LA) having different effects because of the subtle differences in their structures. Subbaiah et al. [48] reported different patterns of incorporation of LA and different CLAs into phospholipids in cultured Chinese hamster ovary cells and HepG2 cells, with significant incorporation of CLAs into lipid rafts. CLA10,12 but not CLA9,11 increased the cholesterol content of lipid rafts. Another study [49] identified that CLA9,11 altered membrane cholesterol distribution and raft formation in colorectal adenocarcinoma HT-29 cells and that this was linked with decreased nuclear translocation of the epidermal growth factor receptor. The authors concluded that CLA9,11 modified membrane structure altering cell signalling and effects associated with lipid raft modification. These studies suggest that the biological activities of CLAs might be driven by membrane-mediated actions. Modulation of lipid rafts has been suggested to be one mechanism by which *trans* FAs affect inflammation [50,51]. Given the findings of the current study (i.e., differential effects of CLA9,11 and CLA10,12 on inflammatory responses in ECs), it seems important to explore the effects of different CLAs on membranes, including lipid rafts further.

Regarding the limitations of this study, some are inherent with the use of an in vitro model to mimic a (patho)physiological process. Even though it is not possible to confidently extrapolate the observations made in this EC model to what happens in the complexity of interactions resulting in a health outcome (atherosclerosis) in humans, in vitro models are useful for understanding the mechanisms involved in the biological effects of compounds such as FAs. In this case, HUVECs have been widely used as a model to study the inflammatory response of the endothelium and the effects of FAs. These cells respond to inflammatory stimuli in a very similar way to other human ECs, such as those derived from the aorta and coronary arteries [52]. Other limitations of the current work include that proteins of the NFκB signalling pathway, EC membrane properties, and adhesion molecules other than ICAM-1 were not measured. 

On the other hand, the study has several strengths. One of these is the consistent checking of FA concentrations in the culture medium before performing every set of experiments. This is rarely performed but is important when polyunsaturated FAs, especially those with conjugated double bonds, are used because they can be unstable and prone to oxidation. Consistently checking the concentration gave confidence that the desired FA concentrations were being used. Other strengths include the use of low (physiological) concentrations of FAs. FAs were added to the culture medium as non-esterified fatty acids (NEFAs). In healthy humans, the fasting plasma or serum NEFA concentration is typically about 600 μM. Burdge et al. [53] reported the contribution of CLA9,11 and CLA10,12 to the plasma NEFA pool in healthy young men prior to and after using supplements of CLA9,11 and CLA10,12. They showed that CLA9,11 contributes on average 0.35% of FAs in the fasting NEFA pool. This would be an approximate concentration of 2 μM. After supplementation with 2.4 g CLA9,11 daily for 8 weeks, the contribution increased to 0.69%, which would be an approximate concentration of 4 μM. Hence, the concentrations of 1 and 10 μM CLA9,11 used in the current study are relevant to the human setting. CLA10,12 contributed 0.03% and 0.28% of NEFAs before and after CLA10,12 supplementation, respectively [53]. Those contributions would be approximate concentrations of 0.2 and 1.7 μM, respectively, so again the concentrations of CLA10,12 used in the current study have some relevance to the human setting. Most other in vitro studies of CLAs use much higher concentrations than those used in the current study (e.g., 12.5, 25, and 50 μM in [14]; 50 μM in [17]; 100 μM in [15] these concentrations have less relevance to the human situation. Thus, our findings are an important contribution to the literature since we used more physiological concentrations of CLAs. It is important to note that CLAs also circulate in the bloodstream in an esterified form, such as in triglycerides, phospholipids, and cholesteryl esters [53]. Thus the total plasma or serum concentration of CLAs, especially CLA9,11, is much higher than the concentrations used in the current study. The current study used the MTT assay to allow the identification of FA concentrations at which we were able to induce loss of viability of the ECs being used. Finally, the incorporation of both CLAs being used into the ECs was confirmed; this incorporation was concentration-dependent for both CLAs. 

In conclusion, the results of the current study suggest differential effects of CLA9,11 and CLA10,12 on inflammatory responses of ECs when the fatty acids are used at physiologically relevant concentrations. CLA10,12 showed several pro-inflammatory effects but also displayed some effects that could be interpreted as anti-inflammatory. On the other hand, CLA9,11 had null or anti-inflammatory effects. These findings suggest a complex modulatory role of CLAs in regulating EC inflammation. The exact signalling pathways affected by these two CLA isomers need further exploration.

## 4. Materials and Methods

### 4.1. Endothelial Cell Model

EA.hy926 cells (American Type Culture Collection, LGC Standards, Teddington, UK) were cultured in high glucose Dulbecco’s Modified Eagle Medium (DMEM) supplemented with 10% foetal bovine serum, 1% L-glutamine-penicillin-streptomycin solution, and 1% HAT (100 µM hypoxanthine, 0.4 µM aminopterin, and 16 µM thymidine); medium and supplements were purchased from Sigma-Aldrich (Gillingham, UK). Cultures were maintained at 37 °C in humidified 95% air and 5% CO_2_. Before their use in experiments, cells were grown in T-175 flasks (Corning TM, Corning, NY, USA) until confluent.

### 4.2. Fatty Acid Treatment

*Cis*-9, *trans*-11 linoleic acid (CLA9,11), *trans*-10, *cis*-12 linoleic acid (CLA10,12), and linoleic acid (LA) (all from Cayman Chemicals, Cambridge, UK) were prepared as 1, 10, and 50 mM stock solutions in 100% ethanol. Before each experiment, the stock solutions were diluted in warm complete culture medium to yield final concentrations of 1, 10, and 50 μM. The corresponding control was a 0.1% ethanol solution diluted in complete medium. For the experiments, EA.hy926 cells were seeded in 96-well plates (for MTT assay, ELISA, and adhesion assay), 6-well plates (for gene expression, flow cytometry), or T25 flasks (for gas chromatography), cultured in complete medium and exposed to different FAs for 48 h. Based on the conditions optimised for studying inflammatory responses of cultured EA.hy926 cells (Figure 8), after the FA exposure period, cells were incubated with TNF-α (1 ng/mL; 20 units/mL) for 6 or 24 h, depending on the assay to be performed.

### 4.3. MTT Assay for Cell Viability

Cell viability was assessed as metabolic integrity using the 3-(4,5-dimethylthiazol-2-yl)-2,5-diphenyltetrazolium bromide (MTT) assay, which measures cellular mitochondrial activity. After the treatments, supernatant was removed and replaced with DMEM containing 0.05 mg/mL MTT (Sigma-Aldrich) (100 µL/well) and samples incubated at 37 °C for 4 h. Supernatants (75 µL) were removed and 75 µL of dimethylsulphoxide (Sigma-Aldrich) added. Absorbance was measured at 540 nm on a plate reader. The effects of FAs and TNF-α on cell viability were normalized to control (i.e., no FA or TNF-α, 0.1% ethanol) cultures (100%).

### 4.4. Gas Chromatography

The FA concentrations and the FA composition of EA.hy926 cells after culture with the FAs of interest were determined using gas chromatography. For FA concentration testing, each FA was diluted in full warm medium from the respective stock in 100% ethanol. For FA incorporation, cells were seeded in T25 flasks (Corning TM, Corning, NY, USA) (5 × 10^5^ cells/mL) and incubated for 48 h with each FA at different concentrations. Afterwards, the cells were inspected under the microscope, scraped off, and counted with a Beckman Coulter cell counter. EA.hy926 cells were resuspended to have 1 × 10^6^ cells/800 µL of 0.9% NaCl solution.

Total lipid was extracted from resuspended cell pellets and culture medium after adding an internal standard (C21:0), using chloroform/methanol (2:1 *v/v*) and NaCl (1 M). Lipid extracts were dried under nitrogen at 40 °C and then resuspended in toluene. FAs were released from the isolated lipids and simultaneously methylated by heating with 2% sulphuric acid in methanol at 50 °C for 2 h. The resulting FA methyl esters (FAMEs) were extracted into hexane and then separated and analysed by gas chromatography using conditions described by Fisk et al. [54]. FAME histograms produced were analysed with Agilent ChemStation software. Thirty-seven FAMEs were used as standard to identify FAs according to retention time and for software calibration. FAs are expressed as µg/10^6^ cells.

### 4.5. Multiplex Magnetic ELISA

Cell culture supernatants were assayed by Human Magnetic Luminex Screening Assay ELISA (R&D Systems, Minneapolis, MN, USA) to measure the concentration of inflammatory factors monocyte chemoattractant protein (MCP)-1, interleukin (IL)-6, IL-8, regulated upon activation, normal T cell expressed and presumably secreted (RANTES), and intercellular adhesion molecule (ICAM)-1. EA.hy926 cells were incubated with the FAs in 96-well flat-bottomed plates (Corning TM, Corning, NY) (1 × 10^4^ cells/100 µL per well) for 48 h and then incubated with TNF-α for a further 24 h. Before the supernatants of each well were collected and stored at −80 °C until analysis, the cells were checked under the microscope. Assays were conducted in accordance with the instructions from the manufacturer. Plates were analysed on a calibrated Bio-Plex 200 analyser using Bio-Plex software (version 6.1, Bio-Rad Laboratories Inc., Berkeley, CA, USA). Lower limits of detection (pg/mL) were IL-6, 1.7; IL-8, 1.8; MCP-1, 9.9; RANTES, 1.8; ICAM-1, 87.9. Due to differences in the ranges of fluorescence values among experiments, the results are presented as % of control.

### 4.6. RNA Isolation, cDNA Synthesis, and Real-Time PCR

Changes in relative gene expression were analysed by RT-PCR. EA.hy926 cells were cultured in 6-well plates (Corning TM, Corning, NY) (cell density of 6 x 10^5^ cells/mL) with FAs for 48 h followed by incubation with TNF-α (1 ng/mL) for 6 h. Taqman Gene Expression Primers (ThermoFisher Scientific, Waltham, MA, USA) were used to determine the expression of *NFκB1* (Hs00765730_m1), *NF**κ**BIA* (Hs00355671_g1), *I**κ**BKB* (Hs00233287_m1), *PTGS2* (Hs00153133_m1), and *IL-6* (HS00985639_m1). Total RNA was extracted from the cells using the ReliaPrep RNA cell Miniprep System (Promega, Southampton, UK). RNA quantity and quality were analysed by NanoDrop. Analysis of RNA using an Agilent Bioanalyzer (RNA Total Eukaryote 2100 Nano) was performed to determine RNA integrity through RIN scores. cDNA was synthesised from total RNA using GoScript Reverse Transcriptase (Promega). Housekeeping reference genes were determined using a geNorm Kit (Primerdesign, Camberley, UK). Quantification of relative gene expression was analysed using YWHAZ, (Hs01122445_g1), CYC1 (Hs00357717_m1), and RPL13A (Hs04194366_g1) as housekeeping genes.

### 4.7. THP-1 Monocyte Adhesion Assay

The adhesion of THP-1 monocytes to EA.hy926 cells was determined using the Vybrant Cell Adhesion Assay Kit (ThermoFisher Scientific). EA.hy926 cells were seeded in 96-well flat-bottomed plates (Corning TM, Corning, NY (density of 2 × 10^5^ cells/mL, 1 × 10^5^ cells per well). After incubation with FAs for 48 h and then with TNF-α for 6 h, calcein-labelled THP-1 cells (5 × 10^4^ cells in 100 µL) were incubated with EA.hy926 cells for 1 h at 37 °C. The co-cultures were carried out in high glucose DMEM supplemented with 10% foetal bovine serum and 1% L-glutamine-penicillin-streptomycin solution. Non-adherent THP-1 cells were removed by gentle washing, 100 µL phosphate-buffered saline (PBS) added to each well and co-cultures read on the Glomax Discover System (Promega). THP-1 monocyte adhesion was measured as a percentage of control (cells incubated with TNF-α, but without prior incubation with FAs, and then with calcein-labelled THP-1 cells). Images of fluorescence-labelled THP-1 monocytes bound to EA.hy926 cells were taken with a Nikon Elipse Ti using NIS elements software (version 4.30, Nikon Instruments, Amsterdam, The Netherlands).

### 4.8. Flow Cytometry

The expression of ICAM-1 (also known as CD54) on the surface of EA.hy926 cells was determined through flow cytometry. EA.hy926 cells were seeded in 6-well plates (Corning TM, Corning, NY, USA) (density of 6 × 10^5^ cells/mL). After incubation with FAs for 48 h and then with TNF-α for 6 h, the cells were detached, centrifuged, and stained with PE-Cy^TM^5-conjugated monoclonal anti-human CD54 (BD Biosciences, San Jose, CA, USA) diluted in staining solution (2% bovine serum albumin in PBS) for 30 min at 4 °C in darkness. Mouse IgG1 κ (PE-Cy^TM^5) isotype was used as a negative control. After staining, cells were analysed by flow cytometry using a FACSCalibur flow cytometer (BD Biosciences). A total of 10,000 events were collected. Percentage of positive cells and median fluorescence intensity (MFI) were measured.

### 4.9. Data Analysis

Data are presented as mean ± SD and were analysed by two-way analysis of variance (ANOVA) or one-way ANOVA followed by Tukey’s post hoc test for pairwise differences. Analyses were performed using GraphPad Prism 6.0. Differences were considered significant when *p* < 0.05.

## Figures and Tables

**Figure 1 ijms-23-06101-f001:**
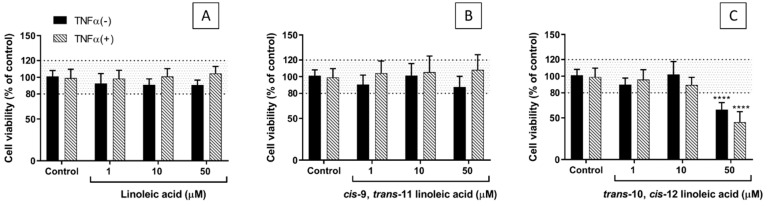
Viability of EA.hy926 cells after preincubation for 48 h with supplemented DMEM containing 0.1% of ethanol (Control) or different concentrations (1, 10, or 50 µM) of linoleic acid (**A**), *cis*-9, *trans*-11 linoleic acid (**B**), *trans*-10, *cis*-12 linoleic acid (**C**), followed by incubation with (+) or without (−) tumour necrosis factor (TNF)-α (1 ng/mL) for 24 h: TNF-α(−) (black bars) refers to cells cultured in the absence of TNF-α while TNF-α(+) (hatched bars) refers to cells cultured with TNF-α. Bars are mean ± SD of 9 samples from 3 experiments. Data were analysed using two-way ANOVA with Tukey’s post hoc test. **** *p* < 0.0001 vs Control.

**Figure 2 ijms-23-06101-f002:**
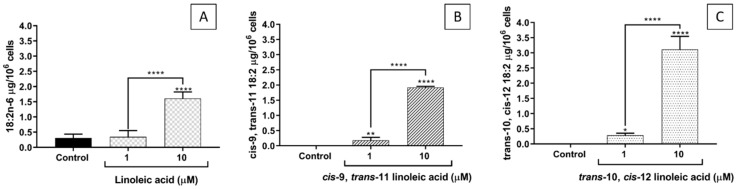
Incorporation of fatty acids into EA.hy926 cells incubated for 48 h with DMEM containing 0.1% of ethanol (Control) or different concentrations (1 or 10 µM) of linoleic acid (**A**), *cis*-9, *trans*-11 linoleic acid (**B**), *trans*-10, *cis*-12 linoleic acid (**C**). Bars are mean ± SD of 6–9 samples from 3 experiments. Data were analysed using one-way ANOVA with Tukey’s post hoc test and represent the total amount of each fatty acid in the cells across all lipids present. * *p* < 0.05; ** *p* < 0.01; **** *p* < 0.0001; where asterisks are shown immediately above a bar they refer to dfference from Control and where asterisks are shown above a horizontal line they refer to differences between the two groups indicated by that line.

**Figure 3 ijms-23-06101-f003:**
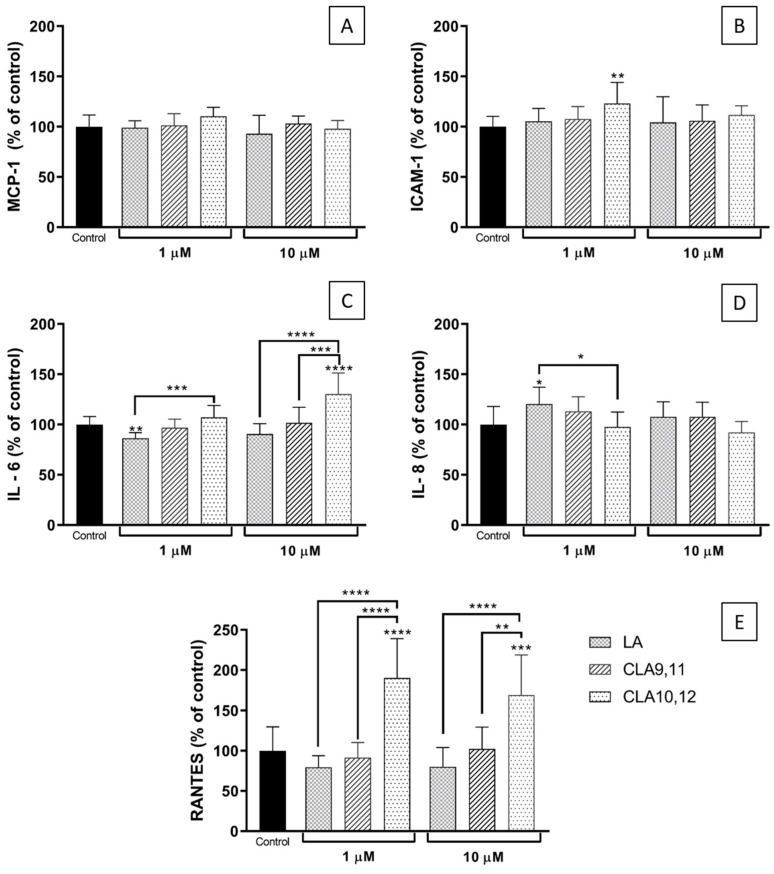
Concentrations (% of control) of MCP-1 (**A**), ICAM-1 (**B**), IL-6 (**C**), IL-8 (**D**), and RANTES (**E**) in the medium of EA.hy926 cells preincubated for 48 h with DMEM containing 0.1% of ethanol (Control) or fatty acid at 1 or 10 µM, followed by incubation with tumour necrosis factor α (1 ng/mL) for 24 h. Bars are mean ± SD of 9 samples from 3 experiments. Data were analysed using one-way ANOVA with Tukey’s post hoc test. * *p* < 0.05; ** *p* < 0.01; *** *p* < 0.001; **** *p* < 0.0001; where asterisks are shown immediately above a bar they refer to dfference from Control and where asterisks are shown above a horizontal line they refer to differences between the two groups indicated by that line. LA, linoleic acid; CLA9,11, *cis*-9, *trans*-11 linoleic acid; CLA10,12, *trans*-10, *cis*-12 linoleic acid.

**Figure 4 ijms-23-06101-f004:**
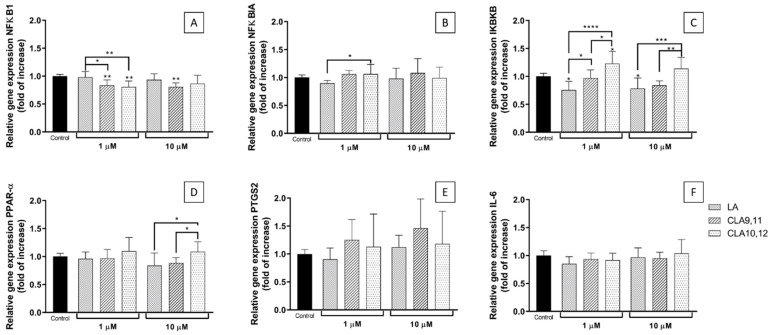
Expression of *NFκB1* (**A**), *NF**κ**BIA* (for IκBα, (**B**)), *I**κ**BKB* (for IκK-β, (**C**)), *PPAR-α* (**D**), *PTGS2* (for COX-2, (**E**)), and *IL-6* (**F**) genes in EA.hy926 cells preincubated for 48 h with 1–10 µM of fatty acid in DMEM containing 0.1% of ethanol (Control) followed by incubation with tumour necrosis factor α (1 ng/mL) for 6 h. Cq values were normalized by the geometric mean of reference targets (RPL13A and CYC1 genes). Bars are mean ± SD of 9 samples from 3 experiments. Data were analysed using one-way ANOVA with Tukey’s post hoc test. * *p* < 0.05, ** *p* < 0.01, *** *p* < 0.001, **** *p* < 0.0001; where asterisks are shown immediately above a bar they refer to dfference from Control and where asterisks are shown above a horizontal line they refer to differences between the two groups indicated by that line. LA, linoleic acid; CLA9,11, *cis*-9, *trans*-11 linoleic acid; CLA10,12, *trans*-10, *cis*-12 linoleic acid.

**Figure 5 ijms-23-06101-f005:**
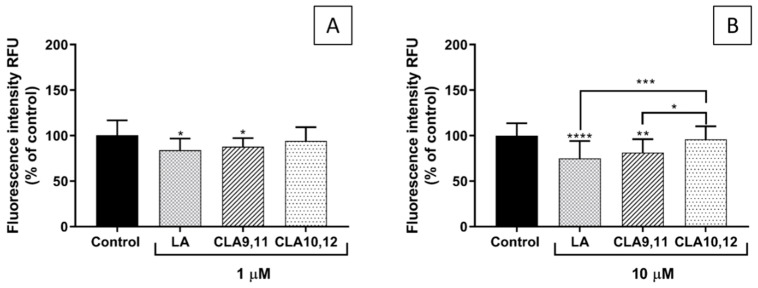
Adhesion of THP-1 cells (% of control) to EA.hy926 cells incubated for 48 h with DMEM containing 0.1% of ethanol (Control) or different concentrations (1 µM (**A**), 10 µM (**B**)) of fatty acid, followed by incubation with tumour necrosis factor α (1 ng/mL) for 24 h and then 1 h co-incubation with THP-1 cells. Bars are mean ± SD of 9 samples from 3 experiments. Data were analysed using one-way ANOVA with Tukey post hoc test. * *p* < 0.05; ** *p* < 0.01; *** *p* <0.001; **** *p* < 0.0001; where asterisks are shown immediately above a bar they refer to dfference from Control and where asterisks are shown above a horizontal line they refer to differences between the two groups indicated by that line. LA, linoleic acid; CLA9,11, *cis*-9, *trans*-11 linoleic acid; CLA10,12, *trans*-10, *cis*-12 linoleic acid.

**Figure 6 ijms-23-06101-f006:**
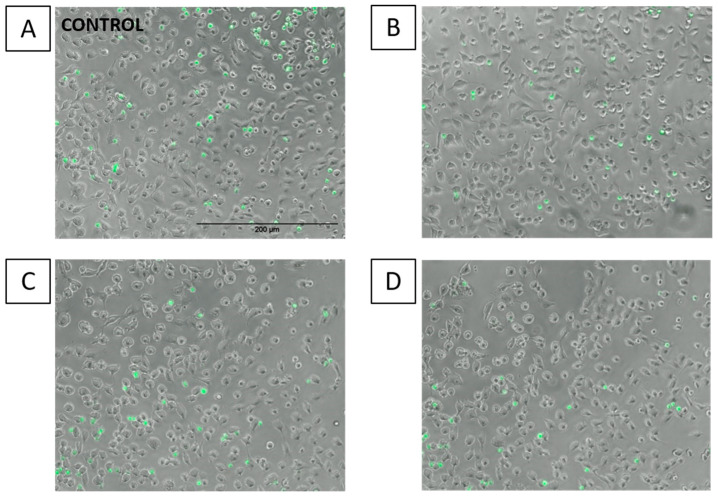
Images of THP-1 cell adhesion to EA.hy926 cells. Adhesion of THP-1 cells to EA.hy926 cells without pre-incubation with fatty acid (control (**A**)) or with 48 h prior exposure to 10 µM linoleic acid (**B**), *cis*-9, *trans*-11 linoleic acid (**C**), *trans*-10, *cis*-12 linoleic acid (**D**), followed by incubation with tumour necrosis factor α (1 ng/mL) for 6 h and then 1 h co-incubation with calcein labelled THP-1 cells. Attached THP-1 cells were visualised by fluorescence microscopy (Nikon Elipse Ti) at a magnification of 100× under transmitted light.

**Figure 7 ijms-23-06101-f007:**
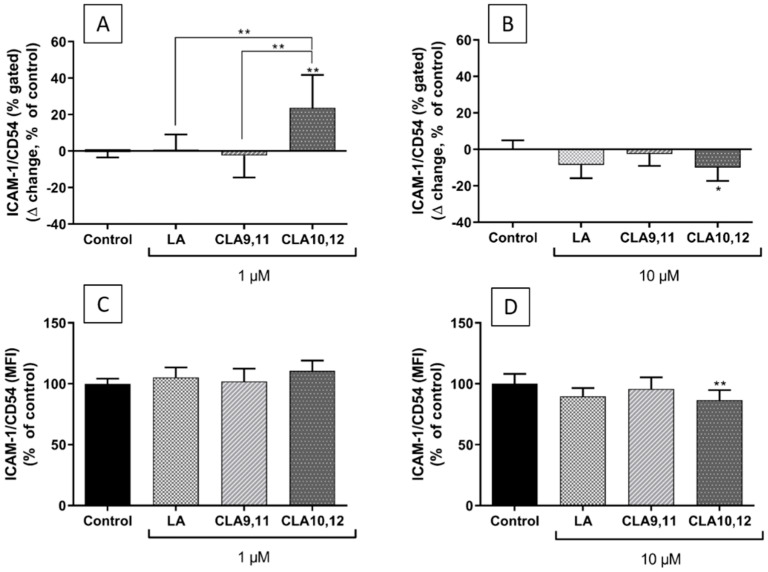
Cell surface expression of ICAM-1 (also known as CD54) as % of EA.hy926 cells gated (∆ change, % of control) (**A**,**B**) and as median fluorescence intensity (MFI, % of control) (**C**,**D**) after preincubation for 48 h with DMEM containing 0.1% ethanol (Control) or different concentrations (1 µM and 10 µM) of fatty acid, followed by incubation with tumour necrosis factor α (1 ng/mL) for 6 h. Bars are mean ± SD of 9 samples from 3 experiments. Data were analysed using one-way ANOVA with Tukey’s post hoc test. * *p* < 0.05; ** *p* < 0.01; where asterisks are shown immediately above a bar they refer to dfference from Control and where asterisks are shown above a horizontal line they refer to differences between the two groups indicated by that line. LA–linoleic acid, CLA9,11–*cis*-9, *trans*-11 linoleic acid, CLA10,12–*trans*-10, *cis*-12 linoleic acid.

**Figure 8 ijms-23-06101-f008:**
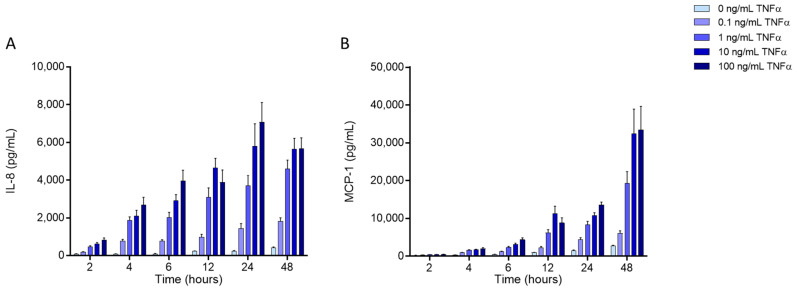
Efect of stimulation of EA.hy926 cells with different concentrations of tumour necrosis factor (TNF)-α on concentrations of interleukin-8 (IL-8; (**A**)) and monocyte chemoattractant protein-1 (MCP-1; (**B**)) in the culture medium. EA.hy926 cells were cultured with different concentrations of TNF-α (0.1 to 100 ng/mL) for different times (2, 4, 6, 12, 24, 48 h). Concentrations of IL-8 and MCP-1 were measured by ELISA (see Section 4.5). Bars are mean ± SD of 3 replicates.

## Data Availability

Data can be made available by contacting the corresponding author.

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
