# Peer review of "Differential Inflammatory Responses in Cultured Endothelial Cells Exposed to Two Conjugated Linoleic Acids (CLAs) under a Pro-Inflammatory Condition"

_ijms, 2022, doi:10.3390/ijms23116101_

Round 1

Reviewer 1 Report

The manuscript titled “Differential inflammatory responses in cultured endothelial cells exposed to two conjugated linoleic acids ( CLAs ) under a pro-inflammatory condition” demonstrated that CLA with trans-10, cis-12 double bonds (CLA10,12) showed pro-inflammatory effects on epithelial cells (ECs) at close-to physiological fatty acid concentrations. On the other hand, CLA with cis-9, trans-11 double bonds (CLA9,11) was demonstrated to have some anti-inflammatory effects on ECs. At 1 to 10 uM concentration, authors showed that 48 hours of CLA9,11 and CLA10,12 treatment did not induce mitochondrial defects. It is demonstrated that CLA9,11 and CLA10,12 were significantly incorporated to cells after 48-hour treatment, in particular at 10 uM concentrations. Authors argued that CLA10,12 treatment induced the secretion of ICAM-1, IL-6, and RANTES, tested by ELISA-based quantifications. Also, the authors argued that CLA10,12 treatment increased IKBKB and PPAR-a transcript expressions, tested by qPCR methods. In contrast, it is argued that NFKB1 expression was reduced by CLA9,11 or CLA10,12 treatment. In addition, the authors demonstrated that linoleic acid and CLA9,11 treatment slightly reduced the recruitment of THP-1 cells to ECs, tested by cell adhesion assay kit and fluorescence microscopy. Last, the authors argued that CLA10,12 slightly reduced ICAM-1 surface expression, tested by FACS.

The authors performed solid experiments with proper controls and methodologies. As the authors pointed out, I appreciate the research using relatively close-to physiological levels of fatty acid concentrations in the human body. Also, it was an excellent experiment to test cell viability after fatty acid treatment as it is well-known that too many free fatty acids may give cells detrimental effects, including oxidative stress. Moreover, finding out that exogenous lipids were incorporated into cells makes a clear mechanistic connection between the lipid and phenotypes.

I have a few comments that can improve the manuscript before publication.

Major comments:

  1. Current research argues that CLA10,12 has pro-inflammatory effects on ECs, but the authors comment (in the introduction and in a very long discussion) on the inconsistency of previous research on the effect of CLA. It is not clear if this particular research has superiority over previous research or if there is a proper explanation for why current research stands and others do not. Please make the argument clear.
  2. From line 289, I expect to find out a mechanism of action of CLA as a part of membrane lipids. Due to its two double bonds, linoleic acid contributes to higher membrane fluidity, although it is not clear how such biophysical properties affect cell signaling. The authors ignore any physical mechanism of CLAs to evoke any downstream phenotypes. Please provide proper information on how CLA, not linoleic acid or other fatty acids, induces the downstream genetic/cellular changes.
  3. As the authors pointed out, I think testing if CLA treatment changes VCAM-1 secretion or expression on the cell surface is valuable information to add to the manuscript. Please experiment to test the involvement of VCAM-1 in CLA-associated phenotypes.
  4. Comparison between different conditions should present variability in standard deviation forms. Please remove SEM and provide SD as error bars in all bar graphs.

Minor comments:

  1. For experiments included in Figures 1, 3, and 5, it is not clear how the authors normalized the data to cell numbers. Cell culture for 48-hour growth or any treatment is expected to generate a significant variation in cell number per well in 96-well plates. Thus, MTT assay, ELISA, and cell adhesion assay should be thoroughly normalized to the final cell concentration of each well. If the data are normalized to the cell concentration properly, please explain how the data are normalized.
  2. Please make it clear that in Figure 2, what the authors measure is FAME from all lipids, not only from free CLA.

Author Response

Thank you for your supportive comments.

In response to your major comments: 

  1. We have tried to better clarify this point in the Discussion (line 391-393).
  2. Of course, the reviewer is correct that the membrane effects of fatty acids are important to their mechanism of action. We did not assess membrane physical properties in this study, but we have now added a section about this in the Discussion (lines 323-359).

  3. We regret that we are not able to do this at the moment because we have no samples left to make these measurements on and the researchers who did this research have now left.

  4. Figure 1 showed SD, although this was incorrectly described as SEM. Other figures have been modified to show SD instead of SEM.

Minor comments: 

4. These cells proliferate slowly and so large changes in cell number during these culture periods are not expected. Figure 1 shows % of all cells that were viable, irrespective of cell number. The unit (%) is not affected by cell number and so normalisation for cell number is not necessary or done. Figure 3 shows cytokine concentrations in the culture medium; these were measured as absolute concentrations and then normalised to the concentrations measured in the medium of control cells (TNF-stimulated but no fatty acids). It is not usual practice to normalise these concentrations to cell number when an identical number of cells has been plated initially as we did here. The fact that some mediators are significantly different from control whereas others are not is an indication that the differences reflect differences in cellular activity not cell number. Figure 5 shows adhesion of calcein-labelled THP-1 cells to ECs again expressed normalised to binding to control cells (TNF-stimulated but no fatty acids). Visual examination by microscopy did not suggest any marked differences in EC number at the time of THP-1 addition to the cultures.

5. We have clarified this in the figure legend.

Reviewer 2 Report

This is an interesting paper whether conjugated linoleic acid (CLA) isomers might possess anti-atherosclerotic properties, which may be related to the downregulation of inflammatory pathways in different cell types, including endothelial cells. The study is well designed and performed and the paper is well written. However, there are some questions remaining which should be answered.

  • In addition to reference 13 a recent review about atherosclerosis should be quoted:

Jebari-Benslaiman S, Galicia-García U, Larrea-Sebal A, Olaetxea JR, Alloza I, Vandenbroeck K, Benito-Vicente A, Martín C. Pathophysiology of Atherosclerosis. Int J Mol Sci. 2022 Mar 20;23(6):3346. doi: 10.3390/ijms23063346.

  • Why hy926 cells and not HUVEC were used?
  • What exactly means TNF-a(-) and TNF-a(+) control?

This should be described in the para “Endothelial Cell Model” in the chapter “Material and Methods”

  • What kind of culture plates were used and how much endothelial cells were seeded initially?
  • Why 10% fetal bovine serum - a quite high concentration – was used? What was the cultivation time?
  • What about the proliferation of the cells. Did you measure a growth curve?
  • Why such a low concentration of TNF-a of 1 ng/ml was used? To induce an inflammatory activation of HUVEC, concentrations of around 200 ng/ml often are used: see e.g.

Schulz C, Krüger-Genge A, Lendlein A, Küpper JH, Jung F. Potential Effects of Nonadherent on Adherent Human Umbilical Venous Endothelial Cells in Cell Culture. Int J Mol Sci. 2021 Feb 2;22(3):1493. doi: 10.3390/ijms22031493.

  • Has ethanol an own effect on hy926 cells?
  • Which medium was used for the coculture system?
  • The MTT test does not describe toxicity but describes the metabolic activity or the viability of cells.
  • The endothelial cell density in images A, B, C, D in Figure 6 differs with the lowest density for controls. Why?

Author Response

Thank you for your comments.

In response to your comments: 

  1. We have replaced the existing Ref 12 with the suggested reference.
  2. EA.hy926 cells are a “type” of HUVEC. They display the properties of HUVECs and give highly reproducible responses in cell culture. They are widely used. A PubMed search on “EA.hy926 cell” gives 1058 publications (747 in the last 10 years).
  3. We only use this terminology in Figure 1. TNF-a(-) is cells cultured without TNF-a and TNF-a(+) is cells cultured with TNF-a. We have made this clearer in the legend.

In response to your comments about the "endothelial cell model":

  1. We have now added details of the plates used in the relevant sections. Cell numbers for most experiments were already included (e.g. lines 449, 469, 499, 501) but are now added where they were not (line 480).
  2. ATCC who provide the cells recommend use of 10% FBS in their product information sheet. Incubation time is already given for each experiment in both the M&M and the figure legends.

  3. These cells proliferate quite slowly. We did not measure growth under the different conditions used.

  4. 1 ng/ml gives good stimulation of these cells with capacity to increase or decrease this according to other exposures (like fatty acids). We have added an additional Figure (Figure 8) to demonstrate this. The paper you mention does use 200 ng/ml but our Figure 8 shows that maximal stimulation is achieved with 10 ng/ml and that 1 ng/ml generates good responses.

  5. Ethanol at 0.1% has no effect on any outcome reported here. This was systematically checked. We have added a statement about this in the Discussion (line 223-226).

  6. We have now added this information (line 500-501).

  7. The reviewer is correct about this. We used the term viability throughout the manuscript except on one occasion in the Discussion where toxicity was used inadvertently; we have corrected this (line 397).

  8. The images that were selected were from different experiments. We have replaced the image for (A) with the control image for those shown in (B), (C), (D).

Round 2

Reviewer 1 Report

Thank you for the responses. I am satisfied with authors' responses. The data and the limits were presented clearly.